# One-Year Seismic Survey of the Tectonic CO_2_-Rich Site of Mefite d’Ansanto (Southern Italy): Preliminary Insights in the Seismic Noise Wavefield

**DOI:** 10.3390/s23031630

**Published:** 2023-02-02

**Authors:** Simona Morabito, Paola Cusano, Danilo Galluzzo, Guido Gaudiosi, Lucia Nardone, Pierdomenico Del Gaudio, Anna Gervasi, Mario La Rocca, Girolamo Milano, Simona Petrosino, Luciano Zuccarello, Roberto Manzo, Ciro Buonocunto, Francesca Di Luccio

**Affiliations:** 1Sezione di Napoli—Osservatorio Vesuviano, Istituto Nazionale di Geofisica e Vulcanologia, 80124 Naples, Italy; 2Osservatorio Nazionale Terremoti, Istituto Nazionale di Geofisica e Vulcanologia, 87036 Rende, Italy; 3Dipartimento di Biologia, Ecologia e Scienze della Terra, Università della Calabria, 87036 Rende, Italy; 4Sezione di Pisa, Istituto Nazionale di Geofisica e Vulcanologia, 56125 Pisa, Italy; 5Dipartimento di Scienze della Terra, dell’Ambiente e delle Risorse, Università degli Studi Napoli, 80126 Naples, Italy; 6Sezione Roma1, Istituto Nazionale di Geofisica e Vulcanologia, 00143 Rome, Italy

**Keywords:** seismic survey, spectral analysis, array analysis, tectonic CO_2_-rich site, Mefite d’Ansanto

## Abstract

A passive seismic experiment is carried out at the non-volcanic highly degassing site of Mefite d’Ansanto located at the northern tip of the Irpinia region (southern Italy), where the 1980 M_S_ 6.9 destructive earthquake occurred. Between 2020 and 2021, background seismic noise was recorded by deploying a broadband seismic station and a seismic array composed of seven 1 Hz three-component sensors. Using two different array configurations, we were allowed to explore in detail the 1–20 Hz frequency band of the seismic noise wavefield as well as Rayleigh wave phase velocities in the 400–800 m/s range. Spectral analyses and array techniques were applied to one year of data showing that the frequency content of the signal is very stable in time. High frequency peaks are likely linked to the emission source, whereas at low frequencies seismic noise is clearly correlated to meteorological parameters. The results of this study show that small aperture seismic arrays probe the subsurface of tectonic CO_2_-rich emission areas and contribute to the understanding of the link between fluid circulation and seismogenesis in seismically active regions.

## 1. Introduction

In the Mefite valley of the Irpinia region (southern Italy, Figure 1 and Figure 2), the existence of large and lethal CO_2_ emissions has aroused interest since ancient times [1]. The Mefite gas emission site was mentioned by Mörner and Etiope [2] in a review paper on the worldwide non-volcanic CO_2_ gas escaping from the upper mantle. The authors highlighted how carbon dioxide measured in non-volcanic “colder” environments, such as Mefite, is much greater than previously assumed and how it significantly affects the global CO_2_ amount. Chiodini et al. [3] estimated a total CO_2_ flux of ∼2000 tons per day at Mefite that is the largest non-volcanic gas emission measured on the Earth. They also found that, under low-wind conditions, the gas flows along a narrow natural channel, producing a persistent invisible and lethal gas river. Roberts et al. [4] considered that the CO_2_ released at Mefite could originate from a source in the underlying anticline in proximity of faults with high permeability able to favor gas migration, providing efficient fluid pathways into the overburden. Giustini and Brilli [5] estimated a reservoir fluid temperature up to 120–125 °C at the depth of about 3000 m below the ground level.

Mefite is set in the Ansanto Valley between the Sannio (to the north) and Irpinia seismogenic regions, at the northern tip of the Irpinia fault system that is associated with the destructive M_S_ 6.9, 1980 Irpinia earthquake [8,9]. This zone falls in the axial part of the southern Apennine that, since the middle-late Pleistocene, experienced a SW-NE extension that determined the formation NW–SE striking dip-slip normal faults [10,11]. The gas leakage at Mefite is probably linked to the presence of these active fault systems that are responsible for the large historical earthquakes in the region [3,7].

The Mefite degassing area lies on the western flank of the structural high of the Frigento Antiform (FA, Figure 1), where calcareous-siliciclastic and marly deposits of the Lagonegro Units crop out [12,13]. At depth, well and seismic profile data show that Miocene siliciclastic deposits of the Lagonegro Units tectonically cover the Cretaceous Apulian Platform carbonates [14,15] that can be found at shallow depths (top at 1128 m depth, Mt. Forcuso 1 well [16]). The area is characterized by the presence of vents and small boiling mud lakes that emit gases, among which CO_2_ is the most abundant [3]. The main emission area (Figure 2b) is at the foot of a landslide deposit, along the south-western flank of a step-sloping hill. The surroundings are sparsely inhabited, with prevailing cultivated fields and several wind parks (the closest at about 1 km towards north), whereas the vegetation is absent or heavily damaged in proximity of the degassing area. The main fluid emission center consists of a bubbling mud pool, the Gray Lake (Figure 2b). From the emission site, the gas flows due to its density and channelizes at the bottom of a narrow valley to the west [5].

The isotopic signature of the CO_2_ testifies a deep origin of the gases emitted at Mefite, whose source is probably the mantle wedge beneath the Apennines along the Tyrrhenian side [3]. Deeply generated fluids may ascent through an interconnected network of fractures and faults and then they may be trapped in crustal pockets, which feed the gas emissions at the surface [3]. The Mefite degassing area is likely fed by the CO_2_ reservoir of Mt. Forcuso 1 well, at about 2 km east of Mefite, that is hosted at the top of the Apulian Platform carbonates and sealed by clayey and calcareous marly sediments of Lagonegrese Units [3,7].

The role of CO_2_ emissions in seismicity occurrence has been discussed by a large number of studies worldwide. Cusano et al. [17] investigated the source properties of long-period earthquakes recorded at Campi Flegrei (Italy) during the 2005–2006 mini-uplift episode, which was quickly followed by CO_2_ outflux increase. A locally fracturing process was hypothesized to increase the medium permeability enhancing fluid migration, which in turn generates earthquakes and a higher fluid flux at surface. Chiodini et al. [18] analyzed repeated seismic sequences in 2009–2018 in central Apennine (Italy) and records of CO_2_ flux whose rate varies accordingly to the seismicity occurrence. Those authors retrieved a model in which the CO_2_ flux modulates the analyzed seismic sequences and highlighted how the CO_2_-enriched slab-derived fluids ascend in the crust and interact with Apennine chain aquifers; in tectonically active zones, the fluid ascent is likely favored by crust fracturing during earthquakes and the pressurized rising fluids can trigger earthquakes. Yamada et al. [19] analyzed multidisciplinary data at Kusatsu-Shirane volcano (Japan), relative to the eruption of 2018, where the onset of surface activity was preceded by volcanic tremor, within a frequency band of 5–20 Hz. Those authors hypothesized a link between the observed seismicity and shear fracture mechanisms induced by sudden hydrothermal fluid injection, containing CO_2_ [20]. In a recent review paper, Di Luccio et al. [9] presented an updated overview of the fluids (CO_2_) and seismicity relationships in the Apennines mountain range (Italy) through a multidisciplinary approach. Those authors highlight the necessity of multiparametric monitoring and long-term analysis of different observables to formulate a reliable conceptual model on the role of fluids in the preparatory phase of earthquakes in the southern Apennines.

A few recent seismological studies have focused on the Mefite area. Pischiutta et al. [7] applied a seismic noise polarization analysis and a single station horizontal-to-vertical spectral ratio method to study the seismic wavefield. They performed 25 measurements of 20 min duration, within 5 km distance from the main emissions and found that the ground motion was significantly polarized in the horizontal plane with a N115 predominant trend in the gas emission area. This polarization vanished moving away from the vents, and Pischiutta et al. [7] explained this directional effect as due to fault-induced fractures. Panebianco et al. [21] tested an automated machine learning technique in the 6–20 Hz frequency band to retrieve tremor signals from the background noise recorded from 30 October to 2 November 2019. Both these studies are based on a short-term dataset.

FURTHER (the role of FlUids in the pReparaTory pHase of EaRthquakes in Southern Apennines, https://progetti.ingv.it/en/further, last access 28 November 2022) is a multidisciplinary Departmental Strategic Project of the Istituto Nazionale di Geofisica e Vulcanologia (INGV) started in 2020 to investigate the involvement of fluids in the seismogenesis of the southern Apennines, Italy. One of the goals of the project is to detect the footprint of fluid contribution to the seismogenic process in the seismic signal recorded at Mefite d’Ansanto [22]. In the present work, we describe the seismic survey carried out at Mefite from September 2020 to September 2021 and present a first insight in the local seismic wavefield properties throughout the inspection of the temporal patterns and the spectral characteristics of the continuous seismic noise signals. Moreover, we show some preliminary results obtained from the application of array techniques.

## 2. Materials and Methods

### 2.1. Preliminary Test and Broadband Station Installation

Firstly, the Mefite emission site was inspected to find the best location and configuration for the installation of a seismic array, MEfite seismic Array (AME, Figure 2). On 29 September 2020, we performed a background seismic noise recording by using a station (MTST) located at the top of the escarpment that hosts the main vents, at about 60 m from the Gray Lake. MTST was equipped with two sensors, one short period (1 Hz) and one broadband (20 s). The main spectral content of the recorded noise resulted between 1 and 20 Hz.

On 20 November 2020, we installed a seismic station at about 100 m from the Gray Lake to the north in a farmhouse. MEFA station (MEfite Further Station A, Figure 2a, Table 1) telemetered the signals to the server in Naples (INGV), via UMTS. The station recorded continuous signals until 31 March 2021 (see also Appendix A).

### 2.2. Seismic Array: Design and Exploration

Seismic arrays are largely used in hydrothermal environments [23,24,25,26] to investigate the seismic noise wavefield and to track sources of non-impulsive signals, such as volcanic or non-volcanic tremor, since they improve the signal-to-noise ratio.

To design the array at Mefite, we evaluate how different configurations correctly sample the recorded signal by using the Array Transfer Function (ATF). Under the hypothesis of a vertically incident monochromatic plane wavefield, the response of the array is given by the beamforming method [23], which in the horizontal plane (k_x_, k_y_) is expressed as (https://www.geopsy.org/wiki/index.php/Array_signal_processing, last access 22 December 2022):(1)$$ATF\left(k-k_{0}\right)=\left\|\frac{1}{N}\sum_{i=1}^{N}exp\left(-jx_{i}\cdot \left(k-k_{0}\right)\right)\right\|$$
where k_0_ is the true wavenumber vector of a single plane wave, k = (k_x_, k_y_), N is the number of sensors, j is the imaginary unit, and x_i_ is the position vector of the i-th sensor. This function exhibits a central peak in which the value is 1 (k_x_ and k_y_ = 0) and secondary aliasing peaks in which the amplitude is less than or equal to 1.

The phase velocity values can be derived as a function of wavenumber (k) or frequency, and their limits depend on the array geometry. Figure 3a shows the performance of the first array configuration (Figure 2), composed of 7 sensors with inter-sensor distance between 45 and 200 m. The resolution limit, kmin/2, is taken as the radius of the central peak of the ATF measured at the mid-height (0.5). The aliasing limit, kmax, is the lowest k value, greater than kmin/2, obtained at the intersection of ATF with the diagonal line at 0.5 (diagonal black line in Figure 3b), in correspondence of the azimuths with the most restrictive limit (Figure 3c). In Figure 3d, the domain of velocities/frequencies that can be correctly reproduced by the array is the portion of the plane comprised between the continuous black curve and the dashed one. The two parameters, kmin and kmax, are directly related to the geometry of the array in terms of interstation distance and aperture [27].

AME first configuration shown in Figure 2, whose ATF is reported in Figure 3, was appropriate to explore the 1.8–5 Hz frequency band and Rayleigh wave phase velocities in the 480–800 m/s range. This array acquired data from 8 June to 24 August 2021. On 24 August, three stations were moved to the new locations (Figure 2), reducing the interstation distances to better analyze higher frequency signals likely coming from the main emission field. The new array configuration recorded data until 28 September 2021. Figure 4 shows the performance of the second array configuration, which is composed of 7 sensors with inter-sensor distance between 38 and 191 m. This second station arrangement permitted to explore the 2.8–8 Hz frequency band and Rayleigh phase velocities in the 400–630 m/s range.

The technical characteristics of the array are reported in Table 2. AME stations were uninstalled on 28 September 2021.

Array techniques [28] strongly contribute to distinguish one source from another by tracking the incoming wavefield directions (backazimuth estimation), by discriminating if a wavepacket propagatioin is surficial or deep (high or low slowness values, respectively) by establishing if a signal is coherent (well identified source) or incoherent (i.e., diffuse sources such as anthropogenic multiple sources). Among the array techniques, beamforming and high resolution [27] allow one to retrieve the wavefield propagation characteristics throughout the estimation of the semblance in time domain [29] and coherence in frequency domain, and the slowness and backazimuth of the seismic wavefield. These methods are commonly applied over sliding time-windows of signals synchronized among the array stations, and filtered in limited frequency bands.

## 3. Results

### 3.1. MEFA Spectral Analysis and Root Mean Square Temporal Pattern

MEFA station acquired high-quality data allowing one to identify the characteristic frequency bands of the noise wavefield. An example of nighttime recordings is shown in Figure 5a,b, where we report the three components of the ground motion and the same signals high-pass filtered at 0.2 Hz. In Figure 5c,d, we show an example of daytime recordings.

We estimated the Power Spectral Density (PSD) of 1-h-long signals (Figure 6), for the three components of the ground motion over the whole recording period. The spectral content resulted very stable in time and allowed us to identify four main frequency bands that are summarized in Table 3. In general, the horizontal components appear more complex than the vertical ones and the main spectral content is centered at 7–8 Hz. In some cases, a very narrow peak is seen at about 1.4 Hz. MEFA did not show any significant spectral content above 15 Hz, unlike MTST, and this indicates the presence of a source closer to MTST and with a high frequency content that rapidly attenuates.

The temporal pattern of Root Mean Square (RMS) of signal amplitude can reveal variations in the energy release, since the seismic signal squared amplitude is proportional to the energy release [30,31]. The comparison between the noise RMS pattern and meteorological parameter behaviors, particularly rainfall, can highlight some exogenous source features. To highlight the temporal characteristics of the noise, we calculated the RMS of the raw continuous and the filtered signal in the four frequency bands of Table 3. Following Cusano et al. [31], we estimated the RMS of 1-h-long time windows and then we averaged the values of the three components of the motion. In Figure 7, we used a horizontal major tick spacing of 7 days to allow for the fast recognition, if any, of the 7-day periodicity typical of the anthropogenic activities. This periodicity is not evident in our record. We also investigated the 24 h periodicity; therefore, in Figure 7(b.1)–(b.5), we show the comparison between daytime (red line) and nighttime (dark blue line) RMS patterns. In general, the RMS day values are slightly higher than the night ones. Higher are the frequencies; smaller is the difference between night and day patterns. In the blp band, the difference is negligible. A significant link seems to exist with the rainfall pattern (Figure 7(a.6)), especially for the blp band (Figure 7(a.5)). Precipitation data are retrieved from Montemarano meteorological station, located about 11 km far from Mefite (www.agricoltura.regione.campania.it/meteo/agrometeo.htm, last access 30 April 2022).

We calculated a linear fit for the RMS of MEFA to find a likely temporal trend for all considered frequency bands, after excluding outliers above 3 µms^−1^. We retrieved a slight decreasing tendency, which agrees with the rainfall reduction over the analyzed time interval.

### 3.2. Array Data Analysis

In Figure 8a, we show an example of noise recorded by the EW components of the array stations in the first configuration. The spatial distribution of the noise amplitude appears higher at AME4, which is closer to the main CO_2_ emission area, and smaller at increasing distances. This spatial pattern is also evident in Figure 8b, where the RMS of the ground velocity recorded by AME is shown on a logarithmic scale: the RMS mean level lowers as the distance from the vents increases. These observations suggest that the main wavefield components are sourced in proximity of the CO_2_ emissions.

To determine the spectral content of the signal recorded at AME array, we estimated the PSD at each station over the whole acquisition period. Results of this analysis show that:(1)The main spectral content is distributed in the four bands of Table 3 as identified in the previous analyses;(2)The spectral content pattern is stable in time;(3)In general, the spectra of the horizontal components are more complex than the vertical ones;(4)Stations closer to the Gray Lake (Figure 2) show a significantly higher spectral amplitude above 5 Hz.

In Figure 9 we report some examples of the spectra at the array stations. The spectral content in 1–5 Hz frequency band, generally shared by anthropogenic and hydrothermal sources, show a non-trivial spatial pattern evidencing the presence of at least 2 main peaks, at 1.4 and 3.0 Hz. The first sharp peak mostly appears in daytime.

We have begun to analyze the array data by applying the beamforming and high-resolution methods [23] in several frequency bands between 1 and 10 Hz on a 10 s sliding window. Preliminary results seem to confirm that the first array configuration (Figure 3, and Table 2) is more appropriate to study signals in the b1 band, while the second configuration (Figure 4, and Table 2) is more effective for b2, since the interstation distances are shorter. An example of the outcomes obtained so far is displayed in Figure 10, where we show the results of the array analyses performed for one day (2 July) of signals using the array spectral methods at 3 Hz. The top plot shows the spectral amplitude of the analyzed signals, computed as the average of the array stations in a narrow frequency band centered at 3 Hz. The high amplitude of the background signal during day hours clearly includes many short transient signals of artificial origin. The coherence of the seismic wavefield at the array stations (Figure 10, second panel) is characterized by scattered values, with most of them between 0.3 and 0.6. Such low values can be determined by seismic wave trains coming from different directions that make up as an uncoherent wavefield. Sometimes, a higher coherence, up to 0.9, indicates a more coherent wavefield, likely due to the short-term predominance of a well-defined seismic source. The backazimuth of the analyzed signals (third panel in Figure 10) indicates that most of the noise sources are in the northern sector (300–360° and 0–60°). The slowness is also characterized by very scattered values, with most of them between 1 and 2 s/km corresponding to apparent velocities between 0.5 and 1.0 km/s typical of surface waves. These features characterize the first 20 h of the day. After 20 UTC, the results of backazimuth and slowness become very different compared to those in the previous hours. The backazimuth values are in a narrow range centered at about 160° that means SE of the array center. The slowness values vary in a narrow range centered at about 2.2 s/km, corresponding to an apparent velocity of about 0.45 km/s. The direction SE from the array center corresponds to the Mefite emission field (Figure 2). Therefore, results of the array analyses indicate that the background seismic signal at 3 Hz contains a contribution likely radiated from the hydrothermal system. However, such hydrothermal tremor has very low amplitude; thus, it is overcome by anthropogenic noise during the daytime. This is confirmed by the much smaller amplitude of the analyzed signals during night hours, from 20 to 24 of 2 July, as shown by the spectral amplitude in the top plot of Figure 10.

Unfortunately, the limited space to install the array reduced its maximum aperture; therefore, frequencies below 1 Hz cannot be explored by using array techniques.

## 4. Discussion

To shed light on the possible links between fluid circulation and local seismicity at and around Mefite d’Ansanto, we performed a seismic survey between 2020 and 2021. It was carried out in the framework of the FURTHER project, and mainly consisted in the acquisition of continuous background signals. Our fieldwork explored the north–western portion of the Mefite emission zone, where the ascent to the surface of deeply derived CO_2_ fluids is favored by high permeability fractures and active faults [3]. The seismic noise spectral content clearly displays four frequency bands, 0.2–1 Hz (blp), 1–5 Hz (b1), 5–10 Hz (b2) and 10–15 Hz (b3) (Table 3). The b1, b2 and b3 bands were always identified (Figure 6 and Figure 9), with the higher spectral amplitude in b2 and b3 in proximity of the main vents (Figure 9). This agrees with Panebianco et al. [21], who retrieved similar spectral distribution by the seismic noise analysis over a limited time period of four days. The stations closer to the main vents (AME4, AME5 and MTST) show a significant spectral peak at frequencies higher than 15 Hz (Figure 9). Similar behaviors are consistent with those in Pischiutta et al. [7], indicating that the peculiar spectral pattern tends to disappear away from the gas emission site. Moreover, the higher the frequencies, the lesser the associated signal patterns seem affected by the 24h periodicity (Figure 7). These observations let us to suppose the existence of a seismic source around the main vents with the most energetic components associated to the b2 and b3 bands and above 15 Hz.

In the b1 band, the wavefield seems to be composed of different contributions, none of which appears definitively dominating, whereas the sharp spectral component at 1.4 Hz likely has an anthropogenic origin. Results of the array analyses obtained so far confirm that uncoherent seismic noise is often predominant in this frequency band. A significant seismic signal at 3 Hz is also found and is likely produced by the degassing activity nearby. In hydrothermal environments, spectral peaks possibly generated by the degassing process are mixed with the contribution from different sources, some of them being probably of anthropogenic origin. These are mainly recorded in the b1 band. Exploring the seismic noise properties at Mt. Vesuvius (Italy) by using array analyses, Saccorotti et al. [32] attributed daily changes of the noise spectral power in the 1–4 Hz band to artificial sources but did not exclude the concomitant action of natural sources. Falanga et al. [33], by performing a detailed investigation of the seismic noise at Ischia Island, found three independent seismic signals in the 1–5 Hz band. A dominant and persistent signal was found at 1–2 Hz and demonstrated as being due to the hydrothermal system, its overtone at 2–4 Hz, and a component linked to anthropogenic activities, which shares the same frequencies of the overtone.

Although the very low amplitude of the signal between 1 and 5 Hz does not allow for the unique characterization of the wavefield and source, results from our work motivate further investigation of the background seismicity. A larger number of seismic stations in and around the Mefite emission site would greatly help the comprehension of the hydrothermal tremor by applying array techniques.

Small aperture arrays, with inter-sensor distances such as AME configurations, were deployed for several studies to explore the wavefield characteristics in hydrothermal environments and allowed to explore low-frequency (<10 Hz) wavefields in detail. The investigation of Saccorotti et al. [32] was based on data recorded by an array of 20 stations, with an interstation distance of 40–500 m. Their configuration allowed them to explore the 1–10 Hz frequency band, in which they were able to discriminate three distinct components of the wavefield associated to three different sources. By applying array analysis to data at Campi Flegrei Caldera (Italy), La Rocca and Galluzzo [23] found the presence of nearly continuous coherent signals that include bursts of 10–15 s duration and spectral content in 1–3 Hz. Those authors interpreted the signals as a sequence of low-frequency earthquakes produced by the same source and linked to the hydrothermal activity. Their deployment was composed of 10 short period sensors (1 Hz) with small aperture (400 m).

These small aperture array configurations do not allow for a correct use of array techniques below 1 Hz, which could give more information about the hydrothermal system dynamics. A more extended array was used by Wu et al. [24] in two weeks of 2015 to investigate the shallow hydrothermal system of Old Faithful geyser at Yellowstone (USA). The deployment consisted of 133 three-component geophones, with an interstation spacing of about 50 m and an aperture of about 1 km, and allowed for an exploration, in great spatial detail, of the hydrothermal activity of the study area. The recorded seismic tremor was associated to the hydrothermal fluid dynamics and steam migration at a low frequency (<10 Hz) and to boiling water and bubble collapse at a high frequency (>10 Hz). In addition, a first-order estimation of the fluid content from seismic data was also possible.

In the case of Mefite, the limited area to install the array does not allow for a significant increase in the array maximum aperture, and this also limits the application of array techniques at shallow phenomena (about 300–400 m).

The blp band results unaffected by 24 h periodicity but shows an evident link with rainfall; although, a relation seems to also exist in the other bands. In general, seismic noise in the blp band is related to natural sources, such as the ocean/sea wave climate [34]. In the Irpinia region, Vassallo et al. [35] observed a strong correlation between peaks in the noise PSDs and wind intensity for time windows longer than one day, thus attributing noise variations to meteorological factors. However, in hydrothermal areas, a link between seismicity and rainfall is commonly observed and correlated to phenomena such as rainfall water percolation [36,37,38]. Commonly, the RMS pattern is compared with meteorological parameter behaviors, especially rainfall, to highlight some exogenous source features. In hydrothermal systems, discharged fluids of deep origin could be significantly contaminated by shallow infiltrations, especially meteoric water (rainfall). Di Napoli et al.’s [39] investigations in 2002–2007 at Ischia island (Italy) evidenced how the rainfall influences the dynamics of the hydrothermal system lowering CO_2_ emissions especially during the winter. Rainfall and seismic series from the Soufrière Hills Volcano (Montserrat) were analyzed by Matthews et al. [40] to investigate the exogenous forcing on the volcanic system in 2001–2003. The real-time seismic amplitude pattern, used as a proxy for volcanic activity, indicates that volcanic response to rainfall gradually penetrates and favors long period and hybrid earthquake nucleation. To detect possible hydrological and tidal triggers of hydrothermal and seismic activity at Campi Flegrei caldera (Italy), Petrosino et al. [41] found that the most numerous and energetic earthquake swarm occurrence in 2005–2016 resulted in the wet season, evidencing a strong influence of tidal and hydrological cycles.

Although the relation between seismic noise and meteorological parameters is clear at Mefite site, we have to account for the considerable distance between the seismic stations and the meteorological station (11 km), which makes any interpretation only speculative at this stage. To investigate at what extent the weather conditions affect the seismic signal, in May 2022 we installed an ad hoc meteorological station close to the Mefite vents. This is a Vantage Pro 2 Wireless Davis equipped with an anemometer, a rain collector, a temperature and a humidity sensor, and a Weatherlink datalogger. It is located close to a broadband seismic station installed within the framework of the FURTHER project and will acquire data at least for a dry–wet season cycle.

The identification of the sources inside the noise requires the extension of the already performed analysis to all available data, as well as more detailed analyses, such as the polarization [42] that allows one to track the wavefield components inside restricted frequency bands. Unfortunately, no CO_2_ flux measures were performed [9] during the array functioning period, which would have permitted to directly associate the seismic wavefield components to the degassing activity.

## 5. Conclusions

In this paper, we describe the seismic survey that was carried out at the non-volcanic CO_2_ degassing Mefite d’Ansanto site located at the northern tip of the Irpinia extensional fault system (M_S_ 6.9) in southern Italy, and we give a first insight in the local seismic noise wavefield by computing RMS, spectral and array analyses. Although the results are preliminary, they reveal interesting features of the noise amplitude at frequencies <5 and >5 Hz that motivate further studies for a better understanding of the dynamics of the reservoir as well as the involvement of fluids in the seismogenic processes on a regional scale. The clear link between seismic noise and CO_2_ emissions from a cold and thermal spring in southern Italy points out the importance of investigating in more detail the mutual interaction of seismogenic sources and fluid reservoirs at depth. Seismological analyses prove that accurate and ad hoc studies are required to understand the dynamics of a hydrothermal source such as the one at Mefite that contributes to the natural CO_2_ emission budget.

## Figures and Tables

**Figure 1 sensors-23-01630-f001:**
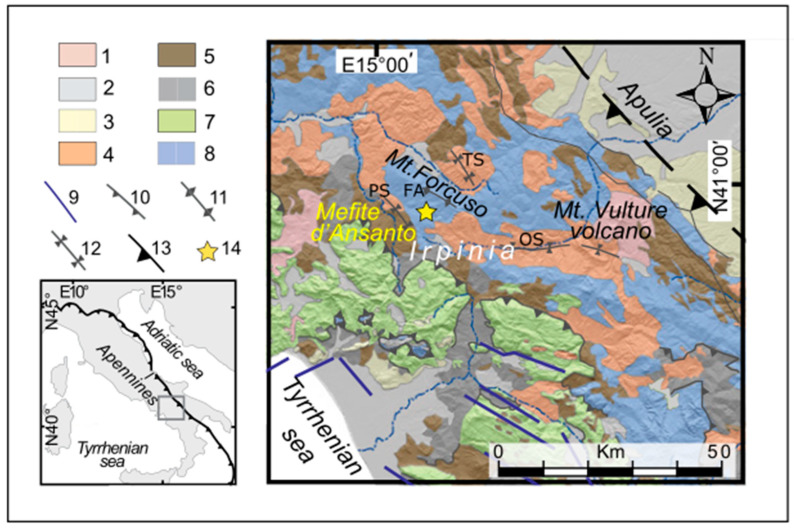
Geological map of the Irpinia region in the southern Apennines, showing the location of Mefite d’Ansanto modified after Ascione et al. [6] and Pischiutta et al. [7]. (1) Middle Pleistocene to Holocene volcanic rocks; (2) lower Middle Pleistocene to present deposits; (3) Lower Pleistocene to lower Middle Pleistocene wedge-top and foreland basin deposits; (4) late Lower Pliocene to Lower Pleistocene wedge-top and foreland basin deposits; (5) Miocene wedge-top and foreland basin deposits; (6) basinal succession (Mesozoic-Tertiary); (7) Apennine Platform carbonates (Mesozoic-Tertiary); (8) Lagonegro—Molise deposits (Mesozoic–Tertiary); (9) main faults; (10) main thrust faults; (11) axis of main antiforms; (12) axis of main synforms; (13) buried thrust front; (14) non-volcanic gas emission. TS, Trevico Synform; PS, Paternopoli Synform; FA, Frigento Antiform; OS, Ofanto Synform.

**Figure 2 sensors-23-01630-f002:**
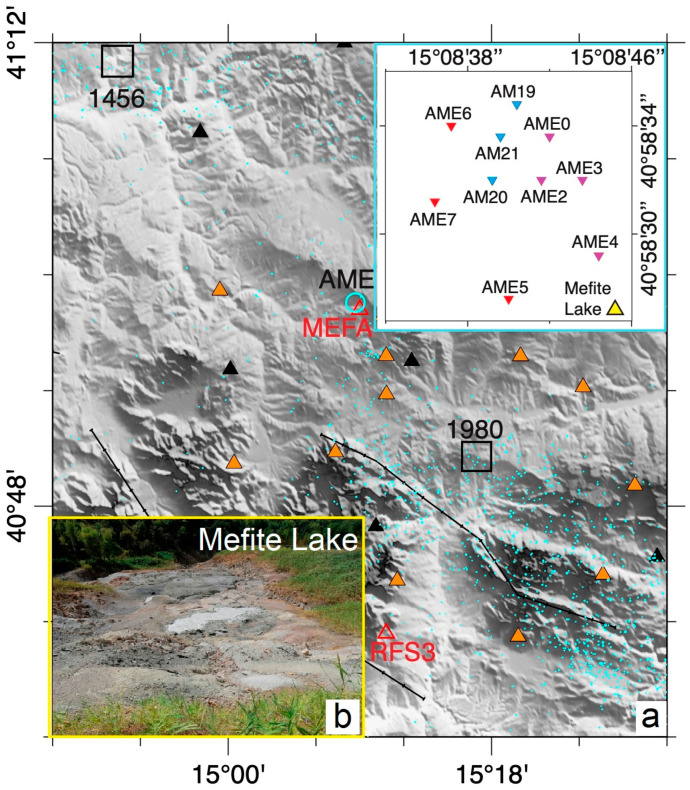
(**a**) Northern Irpinia region (Italy). Orange and black triangles show the locations of the Irpinia and IV seismic networks, respectively. A red triangle shows the position of MEFA station. The open cyan circle indicates the location of the AME array whose station deployment is shown in the inset at the top right. Inset: in this study, two array configurations were used. In the first case the stations indicated by the purple and the red triangles operated from 8 June to 24 August 2021. On 24 August, AME5, AME6 and AME7 were moved to the new locations shown as blue triangles (AM19, AM20 and AM21). The latter configuration (purple and blue triangles) operated until 28 September 2021. The yellow triangle shows the position of “Mefite Lake”. (**b**) A picture of the Mefite main degassing vents, “Mefite Lake” or the “Gray Lake”, taken by one of this paper authors.

**Figure 3 sensors-23-01630-f003:**
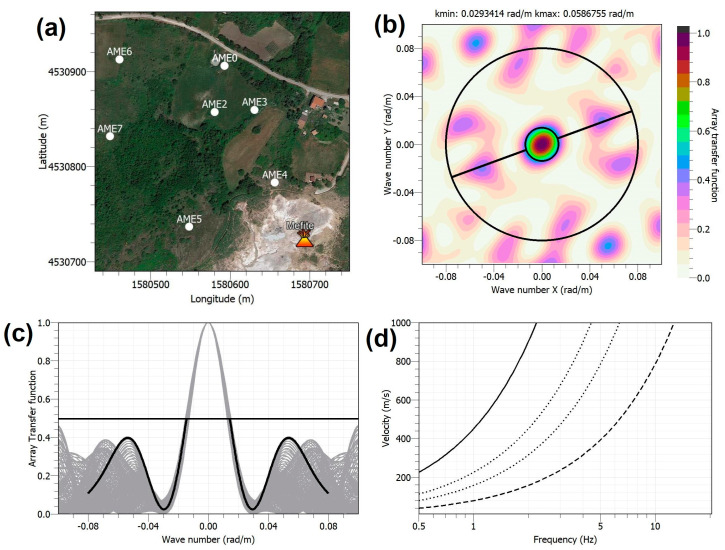
(**a**) AME first configuration. (**b**) The ATF estimated as theoretical frequency-wavenumber response of an array of 7 sensors in the (k_x_, k_y_) plane. The inner black circle corresponds to alias lobe position when the magnitude of the ATF reaches a value of 0.5; this condition occurs along the direction represented by the black line. (**c**) Sections across several azimuths for the theoretical frequency-wavenumber grids of the array (gray curves). The black curve corresponds to the orientation of the black line drawn in (**b**). (**d**) The resolution limits corresponding to (**c**) are shown. Solid and dashed lines represent kmin/2 and kmax, respectively, and dotted lines indicate kmax/2 and kmin.

**Figure 4 sensors-23-01630-f004:**
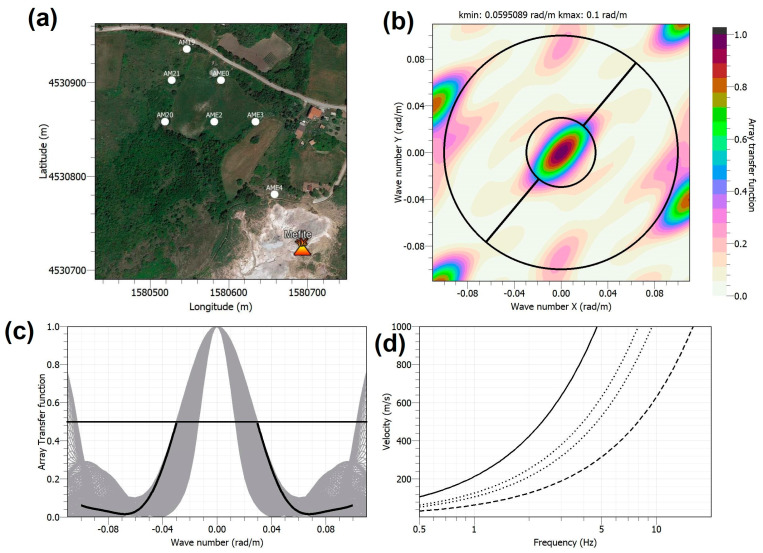
(**a**) AME second configuration. (**b**) The ATF estimated as theoretical frequency-wavenumber response of an array of 7 sensors in the (k_x_, k_y_) plane. The inner black circle corresponds to alias lobe position when the magnitude of the ATF reaches a value of 0.5; this condition occurs along the direction represented by the black line. (**c**) Sections across several azimuths for the theoretical frequency-wavenumber grids of the array (gray curves). The black curve corresponds to the orientation of the black line drawn in (**b**). (**d**) The resolution limits corresponding to (**c**) are shown. Solid and dashed lines represent kmin/2 and kmax, respectively, and dotted lines indicate kmax/2 and kmin.

**Figure 5 sensors-23-01630-f005:**
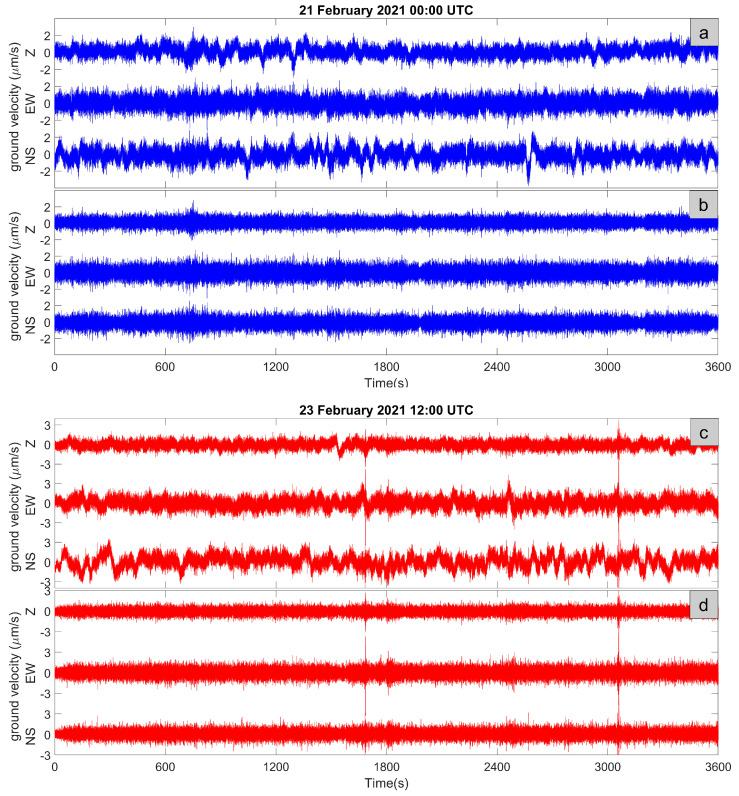
In (**a**,**b**), a nighttime example of the ground velocity recorded by the three components of MEFA station is shown as raw and high-pass filtered signals (0.2 Hz), respectively. The recording starting time is indicated in the plot title. In (**c**,**d**), a daytime example is shown.

**Figure 6 sensors-23-01630-f006:**
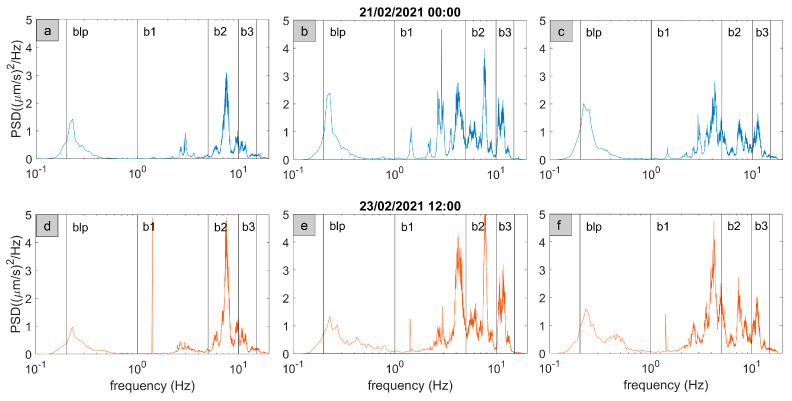
PSDs of the 1-h-long filtered signals reported in Figure 6. (**a**–**c**) show the PSDs of the nighttime signal for the Z, EW and NS components of MEFA station, respectively. PSDs are smoothed. (**d**–**f**) show the daytime PSDs. In each panel, the identified frequency bands (Table 3) are marked by vertical black lines.

**Figure 7 sensors-23-01630-f007:**
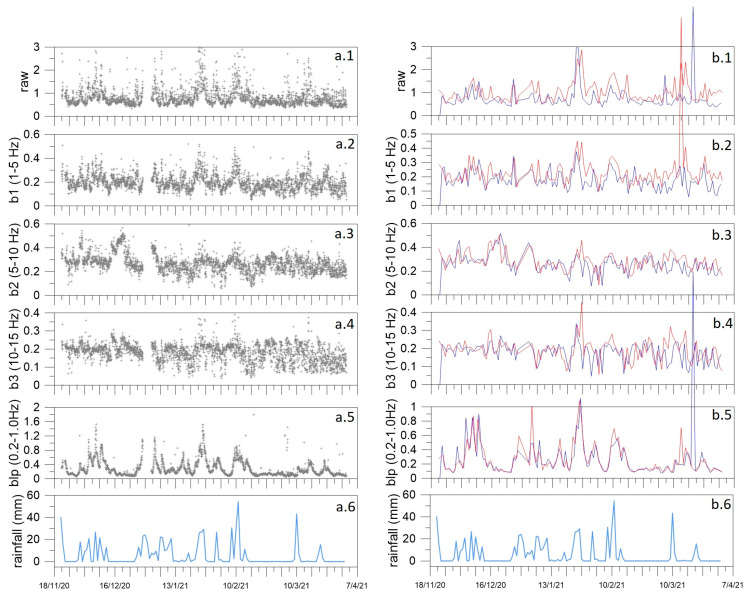
(**a**) From top to bottom, RMS of MEFA signals: raw (**a.1**), filtered in b1 (**a.2**), b2 (**a.3**), b3 (**a.4**) and blp (**a.5**). In plots (**a.1**)–(**a.5**), the gray crosses represent the RMS values estimated over 1-h-long signals and averaged over the three components of the motion. In (**a.6**) the rainfall amount (mm) is plotted (Montemarano). Time on the horizontal axis is expressed in the format day/month/year. (**b**) Day–night RMS comparison. In plots (**b.1**)–(**b.5**), the dark blue lines correspond to the RMS estimated at 00:00 UTC over the whole acquisition period (1 sample-per-day), and the red lines correspond to 12:00 UTC. (**b.6**) is the same plot as (**a.6**).

**Figure 8 sensors-23-01630-f008:**
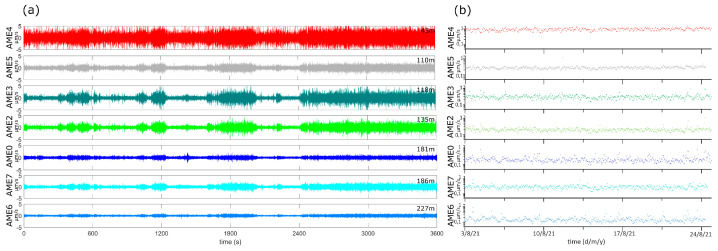
Examples of AME first configuration recordings. (**a**) 1-h of noise signal acquired by the EW components of the array stations. The signal started at 07:00 UTC of 16 June 2021. The vertical units are µm/s (ground velocity). On the right side of each plot we report the distance of the relative station from the center of the lake. (**b**) The RMS estimated for the ground velocity recorded by the AME stations in the period 3–24 August 2021. The vertical axis is logarithmic.

**Figure 9 sensors-23-01630-f009:**
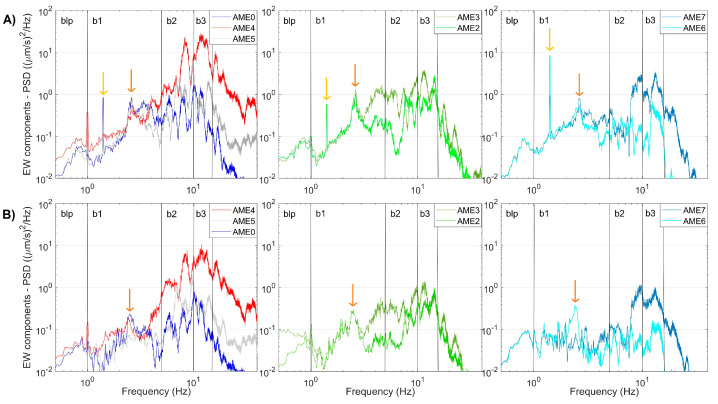
Examples of PSDs estimated for the EW components of the AME stations. The plots refer to 1-h-long signals recorded on 16 June 2021 starting from 07:00 UTC (**A**) and on 10 June 2021 starting at 23:00 UTC (**B**). The vertical black lines mark the blp, b1, b2 and b3 frequency bands (Table 3). The golden and the orange arrows indicate the 1.4 and 3.0 Hz peak, respectively.

**Figure 10 sensors-23-01630-f010:**
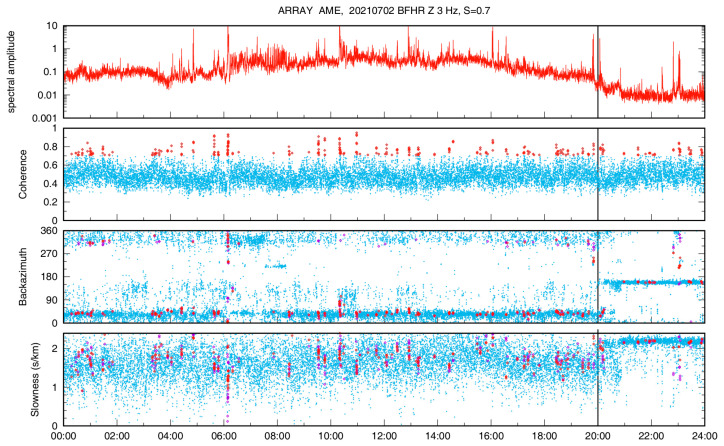
Array analyses at 3 Hz for one day of seismic data recorded by AME on 2 July 2021. Plots from top to bottom show the spectral amplitude, coherence, backazimuth and slowness. Results corresponding to windows with coherence greater than 0.7 are shown by red and magenta symbols for the beamforming and the high resolution techniques, respectively. The black vertical lines mark 20:00 UTC.

**Table 1 sensors-23-01630-t001:** MEFA station specifications.

Name	Latitude (°)	Longitude (°)	Elevation a.s.l. (m)	Sensor Datalogger	Sps (Hz)
MEFA	40.98	15.15	710	Guralp CMG40T-60s Lunitek Atlas	100

**Table 2 sensors-23-01630-t002:** Array coordinates and instrumental characteristics. First configuration: AME0, AME2, AME3, AME4, AME5, AME6 and AME7. Second configuration: AME0, AME2, AME3, AME4, AM19, AM20 and AM21.

Name	Latitude (°)	Longitude (°)	Elevationa.s.l. (m)	SensorDatalogger	Sps(Hz)	Start	End
AME0 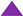	40.9759	15.1450	687	Lennartz 3D-Lite 1 HzReftek	100	8 Jun	28 Sep
AME2 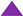	40.9755	15.1449	687	Lennartz 3D-Lite 1 HzReftek	100	8 Jun	28 Sep
AME3 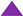	40.9755	15.1454	694	Lennartz 3D-Lite 1 HzReftek	100	8 Jun	28 Sep
AME4 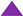	40.9748	15.1456	683	Lennartz 3D-Lite 1 HzReftek	100	8 Jun	28 Sep
AME5 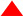	40.9744	15.1445	666	Lennartz 3D-Lite 1 HzTrident–Nanometrics Taurus	100	8 Jun	24 Aug
AME6 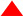	40.9760	15.1438	684	Lennartz 3D-Lite 1 HzNanometrics Taurus	100	8 Jun	24 Aug
AME7 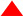	40.9753	15.1436	683	Lennartz 3D-Lite 1 HzTrident–Nanometrics Taurus	100	8 Jun	24 Aug
AM19 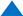	40.9762	15.1446	692	Lennartz 3D-Lite 1 HzNanometrics Taurus	100	24 Aug	28 Sep
AM20 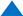	40.9755	15.1443	686	Lennartz 3D-Lite 1 HzTrident–Nanometrics Taurus	100	24 Aug	28 Sep
AM21 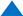	40.9759	15.1444	690	Lennartz 3D-Lite 1 HzNanometrics Taurus	100	24 Aug	28 Sep

**Table 3 sensors-23-01630-t003:** Identified frequency bands.

Band	Frequencies (Hz)
blp	0.2–1
b1	1–5
b2	5–10
b3	10–15

## Data Availability

Data used in this study are acquired in the framework of the 2020–2024 FURTHER project and are available from the corresponding author on reasonable request.

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
