# Peer review of "One-Year Seismic Survey of the Tectonic CO2-Rich Site of Mefite d’Ansanto (Southern Italy): Preliminary Insights in the Seismic Noise Wavefield"

_sensors, 2023, doi:10.3390/s23031630_

Round 1
Reviewer 1 Report
Dear Authors
Introduction:
- The authors focus their study on understanding the connection between the occurrence of earthquakes and the emission of CO2 from geological faults in a non-volcanic region of Italy. They propose an array of detectors which analyze different signals in the range from 1 to 20 Hz. The objective is well established, however the development carried out is very preliminary. They perform a spectral analysis and focus mainly on technical aspects of the array. Considering a brief time, one year, of data, the focus is preliminary tests and installation of the stations, design and exploration. To do this, they represent the earth speed, select four bands and in each one represent Spectral Power Density, indicating that they would have calculated the Root Mean Square.
- Notwithstanding the foregoing, I must point out the following:
- 1.- The article focuses mainly on technical aspects of the proposed arrangement.
- 2.- When it refers to the RMS, it only does so by means of a graphical representation. There is no temporal behavior analysis.
- 3.- No introductory paragraph indicates possible relationships between the occurrence of earthquakes and other phenomena. A priori, a graph of the occurrence of rainfall is presented, without indicating why it is important. How to differentiate the different associated/differentiated sources of seismic noise?
- 4.- With respect to the previous point, only in the discussion is reference made to the results of other authors on possible relationships of other types of events with the occurrence of earthquakes.
- 5.- Nothing is said about CO2 measurements related to the occurrence of tremors. In principle, a reader would assume that there are quantitative results in this regard.
Reviewer 2 Report
The authors conducted a passive seismic experiment at the non-volcanic highly degassing site of Mefite d'Ansanto and found that high-frequency peaks are likely linked to the emission source and low frequencies seismic noise is clearly correlated to meteorological parameters, and small aperture seismic arrays probe the subsurface of tectonic CO2-rich emission areas provided insights to the link between fluid circulation and seismogenesis in seismically active regions.
The manuscript is of good quality and the results obtained are original and important, therefore this manuscript deserves to be published in its current form.
On the form, I have a small problem with the discussion section. Perhaps it would have been more impressive to present previous study results on the small aperture seismic arrays probe studies in other regions and then discuss the differences between this study and previous results including the high-frequency and low-frequencies seismic wavefields at shallow structures.
The scientific question is well established, then it becomes interesting to the community and researchers relative to this topic. The proposed iconography is of good quality and illustrates well the obtained results.
Reviewer 3 Report
The topics of the reviewed article interesting. The results presented indicate the justification for continuing research on the subject.
The introduction is basically sufficient, while it would have been more beneficial if Figure 2, had been clearer. Unfortunately, the color markings are not very clear. No reference in the text to the figure 3c. Citations appropriate to the subject of the article. Methods are presented broadly enough. The results were clearly presented, as reflected in the conclusions indicating the need for further research. This issue was presented in the process of dissemination of project results.
Reviewer 4 Report
Comments for sensors-2149947
This is a well-written paper, which uses the continuously recorded seismic signal to monitor the non-volcanic CO2 degassing at the Mefite d’Ansanto site. This is a very interesting topic, and the analysis is comprehensive, including RMS, spectral, and array analyses. The resulting study shows that high-frequency signals are linked to the CO2 emission source. I have no questions, and a suggestion is for the ambient noise correlation analysis and phase velocity inversion for this array dataset.
Round 2
Reviewer 1 Report
The authors have clearly answered the questions raised in the review I have done. They have clarified methodological aspects and supplemented information on the relationship between occurrences of earthquakes and other phenomena.
Notwithstanding the foregoing, the bulk of the clarifications and complementation of what is required has been based on the results of other researchers. The basis of the conclusions is the description of the detector installation planning for future studies and all the relevant results of the studied phenomenon are always presented as preliminary results.
However, it is a relevant article to be published by scientific development projections. Then I suggest that the title of the article should incorporate the word "preliminary"; that is:
"One-year seismic survey of the tectonic CO2-rich site of
Mefite d'Ansanto (southern Italy): preliminary insights in the seismic noise wavefield".
